# Dielectric Elastomers UV-Cured from Poly(dimethylsiloxane) Solution in Vinyl Acetate

**DOI:** 10.3390/polym12112660

**Published:** 2020-11-11

**Authors:** Seung Koo Park, Meejeong Choi, Dong Wook Kim, Bong Je Park, Eun Jin Shin, Suntak Park, Sungryul Yun

**Affiliations:** 1Human Enhancement & Assistive Technology Research Section, Electronics and Telecommunications Research Institute, 218 Gajeong-ro, Yuseong-gu, Daejeon 34129, Korea; jjeong0527@etri.re.kr (M.C.); bjpark@etri.re.kr (B.J.P.); shin015511@etri.re.kr (E.J.S.); 2Advanced Materials Division, Korea Research Institute of Chemical Technology, 141 Gajeong-ro, Yuseong-gu, Daejeon 34114, Korea; dongwook@krict.re.kr

**Keywords:** elastomer, poly(dimethylsiloxane), photocuring, vinyl acetate, saponification, permittivity

## Abstract

Poly(dimethylsiloxane) (PDMS) has been extensively used as an electroactive polymer material because it exhibits not only excellent moldability but also mechanical properties sufficient enough for electroactive performance despite low dielectric permittivity. Its low dielectric property is due to its molecular non-polarity. Here, we introduce a polar group into a PDMS elastomer by using vinyl acetate (VAc) as a crosslinker to improve the dielectric permittivity. We synthesized a high-molecular weight PDMS copolymer containing vinyl groups, namely poly(dimethylsiloxane-*co*-methylvinylsiloxane) (VPDMS), and prepared several of the VPDMS solutions in VAc. We obtained transparent PDMS films by UV curing of the solution layers. Electromechanical actuation-related physical properties of one of the UV-cured films were almost equivalent to or superior to those of platinum-catalyzed hydrosilylation-cured PDMS films. In addition, saponification of the UV-cured film significantly improved the electrical and mechanical properties (ɛ′ ~ 44.1 pF/m at 10 kHz, E ~ 350 kPa, ɛ ~ 320%). The chemical introduction of VAc into PDMS main chains followed by saponification would offer an efficacious method of enhancing the electroactive properties of PDMS elastomers.

## 1. Introduction

Poly(dimethylsiloxane) (PDMS) is widely known to be a fluid polymer even if it has a high molecular weight because the greater angle and higher length of the Si–O–Si bond in the PDMS main chains in comparison to the C–C–C bond make its chain motion free [1,2]. PDMS can be crosslinked and solidified mainly by a platinum-catalyzed reaction of vinyl groups with hydridosilyl groups in the main chains [2,3]. PDMS shows excellent moldability for those reasons. The vinyl or hydridosilyl groups can be introduced into PDMS by a copolymerization method. The vinyl group-containing PDMS generally has a higher molecular weight than the hydridosilyl group-containing PDMS which acts as a crosslinker. Since the two PDMSs are mechanically mixed together just before shape forming and thermal curing, the platinum-catalyzed reaction cannot uniformly occur during curing, resulting in heterogenous mechanical properties of the PDMS product. Nevertheless, PDMSs have been used as a shapable material for nanoimprint stamps, microfluidic chips, electronic skins, wearable sensors, and actuators owing to their transparency, biocompatibility, and flexibility, in addition to their moldability [4,5,6,7,8,9,10]. However, the other characteristics such as hydrophilicity and dielectric permittivity need to be tuned or improved for effective application of PDMS to the appropriate devices [11,12,13]. PDMS can be modified via polymer reactions or blending methods. These methods show a low degree of efficiency for the preparation of modified PDMS or give rise to deteriorate its own properties such as transparency and flexibility [14,15,16]. In order to avoid disadvantages of the platinum-catalyzed reaction or to form a designed pattern by using a mask or a direct laser, the photo-crosslinking reaction is used to solidify PDMS [6,17,18].

In this study, we tried to enhance the dielectric property of a PDMS film even without deterioration in the other physical properties by substituting the hydridosilyl group-containing PDMS with vinyl acetate (VAc) as a crosslinker. The crosslinking reaction between the vinyl groups in PDMS and VAc is expected upon UV irradiation. Acetate moieties in VAc-grafted or -crosslinked PDMS can be converted to hydroxyl groups after saponification, resulting in improvement of the dielectric permittivity, as well as hydrophilicity, of PDMS. We know that the introduction of hydroxyl groups into nonpolar polymers improves their dielectric properties [19]. First of all, we examined the possibility of forming a film from the vinyl group-containing PDMS/VAc solution layer under UV curing. Further, we evaluated the electrical, mechanical, and optical properties of the UV-cured PDMS films before and after saponification.

## 2. Methods

### 2.1. Synthesis of Poly(dimethylsiloxane-co-methylvinylsiloxane) (VPDMS)

The copolymer was synthesized as shown in Scheme 1. Diethoxydimethylsilane (**1**) and diethoxymethylvinylsilane (**2**) were used as a comonomer. Amounts of 56.93 g (0.384 mol, 80 mol %) of monomer **1** and 15.38 g (0.096 mol, 20 mol %) of monomer **2** were used for the copolymerization. Next, 1.9 mL of 37% hydrochloric acid and 7.5 mL of distilled water were added to the monomers as a catalyst. The detailed polymerization condition, purification method, etc., are shown in our previous report [13]. Yield: 32.9 g (90%). *M*_n_ = 15.0 × 10^4^ g/mol. PDI = 1.66. m:n (mole ratio) = 0.803:0.197. IR ν_max_ (NaCl window, cm^−1^): 1020s, 1093s (Si–O–Si str.); 1261s (Si–CH_3_ str.); 1408m, 1445w (C–H ben., methyl); 1598w (C=C str., vinyl); 2905m, 2963s (C–H str., methyl); 3055w (=C–H str., vinyl). ^1^H NMR δ_h_ (CDCl_3_, 500 MHz): 0.12–0.19 (9H, m, methyl); 5.82–5.87 (H, m, vinyl); 5.96–6.09 (2H, m, vinyl). ^13^C NMR δ_c_ (CDCl_3_, 500 MHz): 1.40–1.99 (m, −CH_3_); 133.44–133.57 (m, =CH_2_); 137.70 (m, −CH=).

### 2.2. Preparation of VPDMS Solutions in Vinyl Acetate and Film Fabrication

After 1.0 mol % of 1-hydroxycyclohexyl phenyl ketone was completely dissolved in vinyl acetate (VAc), the solution was uniformly mixed with VPDMS for 1 day. The VAc concentration was adjusted to ca. 10, 20, and 30 wt %. The polymer solution was placed for a time in order to remove air bubbles formed during agitating. The VPDMS solution was poured on a glass plate and the solution layer was cast with a doctor’s knife. The layer was UV-irradiated under nitrogen for 10 min and dried at 60 °C under vacuum for 4 h to get rid of the unreacted VAc.

### 2.3. Saponification of UV-Cured Film

For preparation of an aqueous saponification solution, 8.0 g of sodium sulfate and 10.0 g of sodium hydroxide were dissolved in 100 g of a mixture of water and methanol (9:1, wt %). After the UV-cured PDMS film was dipped into the saponification solution and treated at 40 °C for 24 h with stirring, it was thoroughly washed with water. Then, the film was vacuum-dried at 60 °C for 4 h.

The other experimental procedures are also explained in detail in the Supporting Information.

## 3. Results and Discussion

We synthesized poly(dimethylsiloxane-*co*-methylvinylsiloxane) (VPDMS) to introduce vinyl groups in poly(dimethylsiloxane) (PDMS) main chains, as shown in Scheme 1. We designed VPDMS to contain more methylvinylsiloxane moieties than the commercial PDMSs (Sylgard 184^TM^, Elastosil P7670^TM^, and VDT 954^TM^) do in order to enhance the possibility of crosslinking, as well as chemically attaching a vinyl- or acrylate-containing compound to the PDMS main chains [6]. We settled the amount of methylvinylsiloxane moiety at ca. 20 mol %. The VPDMS copolymer composition is estimated from the result of ^1^H NMR spectroscopy (Appendix A). We could calculate the copolymer composition by using the peak areas near 0 and 5.8–6.1 ppm. The former is designated to protons in Si–CH_3_, and the latter to protons in the vinyl group. The calculated amount of methylvinylsiloxane moiety in the copolymer was 19.7 mol %, which nearly equals the feeding mole ratio. Even though the copolymer has a high molecular weight (*M*_n_, number-average molecular weight, =15.0 × 10^4^ g/mol and *M*_w_, the weight-average molecular weight, =24.9 × 10^4^ g/mol), the transparent copolymer exhibits a highly viscous liquid state, as expected. The copolymer is well identified in Appendix A.

We obtain homogenous VPDMS solutions in vinyl acetate (VAc) even if the difference of the solubility parameters between PDMS (14.9~15.5 MPa^1/2^) and VAc (18.4 MPa^1/2^) is comparatively large [20,21,22]. However, the solubility parameter value of VAc is calculated from the method of van Krevelen and Hoftyzer. The calculated value of PDMS from the method is 17.5 MPa^1/2^ [22], which is closer to that of VAc. 1-Hydroxycyclohexyl phenyl ketone used as a photoinitiator could be fully activated under our UV power condition (Appendix A) [23]. We optionally prepared three kinds of VPDMS solutions in which the content of VAc was settled at ca. 10, 20, and 30 wt % to identify a suitable condition for film formation during UV curing. Table 1 shows component ratios of the solutions in detail. We anticipate that VAc reacts with the methylvinylsiloxane moiety of VPDMS in the solution layer containing the photoinitiator upon UV irradiation, resulting in VAc-grafted or -crosslinked PDMS, as shown in Scheme 2. Films were well formed after UV curing in all cases, as expected. The film thickness was measured at ca. 200 µm. We know from Figure 1a that VAc was homopolymerized or reacted to VPDMS in the UV-cured films. The absorption peak appears near 1724 cm^−1^ due to the stretching vibration of the carbonyl group in poly(vinyl acetate) (PVAc) and increases in intensity as the VAc content rises. This peak does not come from the unreacted VAc because the VAc would be completely removed during the vacuum drying of the UV-cured film near the boiling point of VAc.

Figure 1b shows the stress–strain curves of the UV-cured films. The films UV-cured from the V20 and V30 solution layers seem quite brittle. This is because more VAc participated in the homopolymerization, as well as in the graft polymerization, to PDMS as the VAc content in the solution increased. Even if the vinyl acetate group is slowly homopolymerized in comparison to acrylate or vinyl groups under UV curing [24], VAc might be more homopolymerized as the VAc content increases, which affects the mechanical properties of the UV-cured films. Since the photoinitiator content in the VPDMS solution increases along with a rise in the VAc content, the vinyl groups of VPDMS react more with each other to result in additional crosslinking between the PDMS chains. This is because a photopolymerization rate is proportionate to the square root of a photoinitiator concentration [25,26]. The film UV-cured from the V10 solution layer was only enough for electromechanical actuation. In this case, VAc could be more grafted to VPDMS rather than be homopolymerized because the mole ratio of vinyl groups of VPDMS is higher than that of vinyl acetate groups in the solution, 0.62. The mechanical properties of the UV-cured film from the V10 solution layer were lower than or nearly equivalent to those of the platinum-catalyzed hydrosilylation-cured PDMS films including the crosslinked films prepared from Sylgard 184^TM^ and Elastosil P7670^TM^ via their suggested recipes (Appendix A) [13]. The detailed property values of all the UV-cured films are summarized in Table 1. Even though VAc is homopolymerized during UV curing, the PVAc homopolymer could be well dispersed in the PDMS matrix due to the steady growth of compatibility between PVAc and PDMS-*g*-PVAc [27]. In addition, the refractive indices of PDMS and PVAc are similar, at 1.43 and 1.47 at a wavelength of 589 nm, respectively [28]. Therefore, the UV-cured films show excellent transparency in a visible region (*T* > 90%), which is almost equal to that of the platinum-catalyzed hydrosilylation-cured PDMS (Figure 1c) [13]. The UV-cured film shows a bathochromic shift as the VAc content increases because the PVAc content increases in the film whether VAc is homopolymerized or grafted.

We saponified the films prepared from the V10 and V30 solution layers. The acetate groups in the UV-cured films would be converted to the hydroxyl groups via hydrolysis, as shown in Scheme 2. An aqueous saponification solution for the heterogenous reaction was prepared according to a previous report [29]. We used a sodium hydroxide and sodium sulfate solution in a mixture of water and methanol for the methanolysis of PVAc in our UV-cured films. The sodium hydroxide acts here as a catalysis for the methanolysis. The use of sulfate ion in the solution is for preventing the homopolymerized poly(vinyl alcohol) (PVA) in the UV-cured film from dissolving out into the saponification medium. For comparison of IR spectroscopy results before and after saponification of the film UV-cured from the V30 solution layer, we used an attenuated total reflection (ATR) method. After saponification of the UV-cured film, the absorption peak near 1724 cm^−1^ completely vanished because of the conversion of acetate groups to hydroxy groups (Figure 2a). We could not obviously find the absorption peak at 3200–3400 cm^−1^ designated to the OH stretching vibration that was expected from the converted PVA. However, we know that the peak in a heterogeneously saponified PVAc film is much less noticeable than that in a PVA film [30]. Therefore, the expected absorption peak in our saponified film may not markedly appear in the IR spectrum acquired especially from the ATR technique. The degree of saponification could not be quantitatively calculated for that reason. Nevertheless, our method of chemically or physically introducing water-soluble PVA into hydrophobic PDMS should be comparatively effective because they cannot be homogenously reacted or blended together due to their big difference in hydrophobicity [10,31].

The thickness of UV-cured films was slightly reduced by ca. 10% after saponification because PVA is denser than PVAc. In addition, the homopolymerized PVA could be partly dissolved out in the saponification solution [29]. The UV-cured film prepared from the V10 solution became tougher after saponification, as shown in Figure 1b. It might be due to the hydrogen interaction between OH groups in the PVAs [32]. The initial modulus (E), maximum stress, and strain (ɛ) values were increased from 220, 215, and 141% to 353 kPa, 757 kPa, and 320%, respectively. Figure 2b exhibits that the storage permittivity (ɛʹ) of the UV-cured film increased from 30.8 to 44.1 pF/m at 10 kHz owing to growth of the dipole moment coming from the OH groups newly formed after saponification. The loss permittivity was also lessened below 0.05 pF/m. E and ɛ’ for the crosslinked films fabricated from Sylgard 184^TM^ and Elastosil P7670^TM^ were measured at 800 and 159 kPa and 26.9 and 25.4 pF/m at 10 kHz, respectively (Appendix A). Only the introduction of VAc into PDMS slightly improves the dielectric property. In the result, the UV-cured film, as well as the saponified film, might be superior to platinum-catalyzed hydrosilylation-cured PDMS films including the commercial PDMS as an electroactive polymer material because the actuation efficiency is proportional to ɛʹ and inversely proportional to E [33,34]. As another indirect evidence of the saponification reaction, we checked water contact angles of the UV-cured film surface before and after saponification. The contact angle was slightly decreased from 106.9° to 102.0° after saponification (Appendix A). The value is known to be ca. 110° for Sylgard 184^TM^ film [16]. The UV-cured film still showed very high transparency and a slight bathochromic shift after saponification (Figure 1c).

## 4. Conclusions

We developed a UV-cured poly(dimethylsiloxane) (PDMS) film with high dielectric and optical properties, as well as a low initial modulus. We introduced a lot of vinyl groups into PDMS and prepared the PDMS solutions in vinyl acetate (VAc) in order to achieve a crosslinked PDMS film containing hydroxyl groups. As designed, films were simply prepared by photocuring of the solution layers and VAc was chemically attached and grafted to PDMS through the curing process. One of the UV-cured PDMS films exhibited a low initial modulus and improved storage permittivity, along with high transparency. Saponification led the acetate groups to convert the hydroxyl groups in the UV-cured film, resulting in a rise in the initial modulus and strain, as well as storage permittivity, of the film. The UV-cured PDMS film, whether saponified or not, is more advantageous than platinum-catalyzed hydrosilylation-cured PDMS films to apply to dielectric elastomer actuators in terms of not only electromechanical actuation but also the fabrication process. In a continuation of our work, we will examine the potentiality of using the UV-cured elastomer as an electroactive polymer.

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
