# Peer review of "Dielectric Elastomers UV-Cured from Poly(dimethylsiloxane) Solution in Vinyl Acetate"

_polymers, 2020, doi:10.3390/polym12112660_

Round 1
Reviewer 1 Report
The paper is well organized and prepared. The research methodology and structure are on very high level. I recommend the article to publication. I have only two minor points to address:
- Please show what is chemical structure of Irgacure 184, and use its chemical name
- It is easier to use ± symbol to indicate the standard deviation of the parameters (Table 1)
Author Response
Reviewer #1: The paper is well organized and prepared. The research methodology and structure are on very high level. I recommend the article to publication. I have only two minor points to address:
- Please show what is chemical structure of Irgacure 184, and use its chemical name
- We substituted the commercial name of photoinitiator, Irgacure 184, with the related chemical name, 1-hydroxycyclohexyl phenyl ketone, at lines 73, 104, and 113. It was corrected in the supporting information, as well. The chemical structure was newly represented in Scheme 2.
- It is easier to use ± symbol to indicate the standard deviation of the parameters (Table 1)
- The ± symbol was introduced in Table 1 as recommended. The old standard deviation-related footnote and the footnote No. (#4) were removed and the existing footnote No. 5 was changed to No. 4.

Reviewer 2 Report
Recommendation: Minor revision
Comments:
- IR data is not very shart. Can you measure the hydroxyl value or saponification value for the bulk polymer?
- Will a blend of PDMS and polyvinylalcohol give the same results? Conduct a control experiment or provide specific literature.
- Correct these:
Line 19: “several of the VPDMS solutions”
Line 88: “more methylvinylsiloxane moieties”
Line 111: “solubility parameters between”
Author Response
Reviewer #2: Recommendation: Minor revision
Comments:
- IR data is not very shart. Can you measure the hydroxyl value or saponification value for the bulk polymer?
- We cannot obtain the degree of saponification from the ATR results because of obscureness of the OH peak. So, the related short discussion was added at line 171.
- Will a blend of PDMS and polyvinylalcohol give the same results? Conduct a control experiment or provide specific literature.
- I think it is impossible to blend PDMS with PVA because they are apparently immiscible. Since PDMS is very hydrophobic and, on the other hand, PVA is hydrophilic, they don’t have cosolvent. As far as I know, there is no polymer blend system like this. So, the related short discussion was added at lines 172-174 and a new reference #31 was added at lines 295-297. Also, three reference numbers were slightly changed at lines 183 and 193 due to introduction of the new reference to the main text.
- Correct these:
- I appreciate the referee’s correction.
Line 19: “several of the VPDMS solutions” - corrected as suggested.
Line 88: “more methylvinylsiloxane moieties” - corrected as suggested.
Line 111: “solubility parameters between” - corrected as suggested at line 110.

Reviewer 3 Report
It's a good work. The writing is properly structured and well written. The methodology is clear and has been adequately developed. The conclusions are perfectly supported. The results obtained are of interest in the field of polymer applications.
Author Response
Reviewer #3: It's a good work. The writing is properly structured and well written. The methodology is clear and has been adequately developed. The conclusions are perfectly supported. The results obtained are of interest in the field of polymer applications.
- I appreciate the referee’s evaluation.
